Interictal neural fragility predicts seizure onset zone and surgical outcomes in drug-resistant epilepsy

Pang Yue 1
Yang Yixuan 2
Lin Yunqing 1
Zhu Jianyu 1
Liu Penghui 1
Tian Yu 1
Wang Feng 1
Mei Zhen 1
Kang Dezhi 1
Cao Miao 2 3 4 miao.cao@pku.edu.cn
Lin Yuanxiang 1 lyx99070@163.com
1 Fujian Medical University , Fuzhou , China
2 Beijing City Key Lab for Medical Physics and Engineering , Beijing , China
3 Swinburne University of Technology , Victoria , Australia
4 Peking University , Beijing , China
Oliveira Sonia
Electronic publication date: 2025 Jul 1
Publication date: 2025
Volume: 13
Electronic Location ID: e19548
Received 2024 Nov 12; Accepted 2025 May 12
Copyright: © 2025 Pang et al.
Copyright year: 2025
Copyright holder: Pang et al.
License: This is an open access article distributed under the terms of the Creative Commons Attribution License, which permits unrestricted use, distribution, reproduction and adaptation in any medium and for any purpose provided that it is properly attributed. For attribution, the original author(s), title, publication source (PeerJ) and either DOI or URL of the article must be cited.
License URL: https://creativecommons.org/licenses/by/4.0/

Keywords: Epilepsy, Drug-resistant epilepsy (DRE), Seizure onset zone (SOZ), Interictal neural fragility, Stereoelectroencephalography (SEEG)

Funding: Innovation of Science and Technology, Fujian Province 2019Y9121 and 2019Y9123 Startup Fund for scientific research, Fujian Medical University 2021QH2031 This work was supported by grants from Joint Funds for the Innovation of Science and Technology, Fujian Province (No. 2019Y9121), Joint Funds for the Innovation of Science and Technology, Fujian Province (No. 2019Y9123) and Startup Fund for scientific research, Fujian Medical University (2021QH2031). The funders had no role in study design, data collection and analysis, decision to publish, or preparation of the manuscript.

==============================
Epilepsy is a chronic neurological disorder affecting approximately 70 million individuals worldwide, with a significant subset of patients exhibiting drug-resistant epilepsy (DRE). Accurate identification of the seizure onset zone (SOZ) is crucial for successful surgical intervention. This study investigates interictal neural fragility as a potential biomarker for predicting SOZ and guiding treatment outcomes in DRE patients. By applying dynamic mode decomposition (DMD) techniques to interictal stereoelectroencephalography (SEEG) data from 30 patients, we generated patient-specific dynamic network models and constructed fragility heatmaps. Our findings demonstrate that patients with favorable surgical outcomes exhibit significantly higher fragility in the SOZ during interictal periods. The fragility-based SOZ prediction model showed high sensitivity and specificity, with a strong concordance between the predicted SOZ and clinically identified treatment targets. This study highlights the clinical utility of interictal neural fragility in enhancing SOZ localization and improving treatment strategies for patients with low seizure frequency. Future research should focus on integrating this model into clinical workflows and exploring its potential in personalized treatment approaches.

Introduction

Epilepsy is a chronic neurological disorder affecting approximately 70 million individuals worldwide. Despite the availability of over 20 antiepileptic drugs (AEDs) for symptomatic management of seizures, approximately one-third of patients exhibit drug resistance, necessitating consideration of surgical resection or neurostimulation treatment strategies in clinical practice (Sander & Shorvon, 1996; Englot & Chang, 2014). In patients with drug-resistant epilepsy (DRE), the success of surgical intervention hinges on the accurate identification of the epileptogenic zone (EZ)—the cortical area responsible for seizure generation—and its relationship to the seizure onset zone (SOZ), the region where abnormal discharges initiate clinically or electrographically (Lüders et al., 2006). While the SOZ (often identified via SEEG) is a critical component of the EZ, emerging evidence suggests the EZ is a network phenomenon, involving distributed nodes and connectivity patterns that sustain epileptogenicity (Bartolomei et al., 2017).

Stereoelectroencephalography (SEEG) has been validated as an effective and relatively safe intracranial imaging technique for SOZ localization. This method relies on the recording and analysis of multiple seizure events (Mullin et al., 2016; Englot, 2018; Bourdillon et al., 2017; Engel, Van Ness & Rasmussen, 1993). However, the process of intracranial monitoring typically extends over several days to weeks to capture the patient’s characteristic or atypical seizures, which inherently limits its diagnostic and prognostic value in patients with infrequent seizures (Jobst et al., 2020). Therefore, there is a pressing need to develop analytical methods for interictal SEEG data that can effectively estimate the SOZ and predict patient outcomes without the requirement for prolonged intracranial recordings, thereby enhancing patient care quality and reducing healthcare costs.

The activity of the brain can often be conceptualized as a network system (Bancaud et al., 1970). Sritharan & Sarma (2014) proposed that epileptic seizures can be viewed as a manifestation of brain network dysfunction, wherein seizures are induced by disrupting network stability. They further suggested that alterations in network function are primarily driven by changes in a small subset of specific network nodes, classified as clusters of vulnerable nodes. Neural fragility serves as a metric for assessing the vulnerability of different nodes; specifically, the lower the interference required for a node to deviate from a stable state and impact the entire network, the more fragile that node is. Subsequent research by Li et al. (2017) demonstrated that neural fragility, as a stable biomarker derived from ictal intracranial EEG data, exhibits robust predictive capability for identifying the SOZ. They observed that post-surgical resection, the fragility of the SOZ during interictal periods significantly decreases (Burns et al., 2014; Yaffe et al., 2015; Bernasconi, 2017; Li et al., 2017). However, whether network fragility derived from interictal data can also serve as a biomarker for predicting the SOZ has not been thoroughly explored, despite its potential implications for advancing the clinical study of focal epilepsy (Jiang et al., 2022; Singh, Sandy & Wiebe, 2015; Breiman, 2001; Lemaître, Nogueira & Aridas, 2017; Li et al., 2017; Chao, Andy & Leo, 2004; Li et al., 2019; Lagarde et al., 2019). In patients undergoing SEEG monitoring, there is an abundance of interictal SEEG data, capturing numerous interictal epileptiform discharges (IEDs), whereas the number of captured and analyzed seizure events is relatively limited. If neural fragility based on interictal SEEG data can serve as an effective biomarker for predicting the SOZ, treatment targets, and patient prognosis, it would not only deepen our understanding of epileptic networks but also provide critical support in clinical decision-making.

This study aims to utilize interictal SEEG data to compute biomarkers for localizing the epileptogenic zone, following these steps: (i) applying dynamic mode decomposition (DMD) techniques to generate patient-specific dynamic network models from interictal SEEG data; (ii) identifying SEEG channels corresponding to the epileptogenic zone, characterized by the fragility of network nodes. We applied this algorithm to snapshots of interictal iEEG data from 30 patients at a single clinical center and assessed the stability and predictive performance of the biomarkers through the following methods: (i) conducting statistical analyses to compare the average fragility of SEEG-identified SOZ and non-SOZ channels; (ii) constructing a predictive model within a random forest framework to identify SOZ channels, and comparing its concordance with SOZ channels identified by clinicians (based on EEG interpretation) and clinically effective treatment targets in patient groups with different outcomes (success and failure); (iii) developing a logistic regression model to analyze the effectiveness of fragility metrics in predicting surgical outcomes, examining their influence relative to other clinical variables, and illustrating their supplementary role in clinical treatment through specific case studies. This study aims to validate the use of interictal SEEG data in predicting the SOZ and surgical outcomes without the need for prolonged intracranial recordings, with the goal of future application in quantitative clinical EEG analysis to improve diagnostic and treatment efficiency while reducing healthcare costs.

Methods

Patient population

This retrospective study included 30 adult patients with drug-resistant epilepsy (DRE) recruited from June 2022 to June 2023 (mean age 25.78 ± 11.37 years (mean ± standard deviation, SD)) who underwent SEEG monitoring followed by surgical intervention. Post-SEEG interventions included radiofrequency thermocoagulation (RFTC, 27 patients), laser interstitial thermal therapy (LITT, five patients), resective surgery (11 patients), and responsive neurostimulation (RNS, two patients). Comprehensive preoperative evaluations were conducted at our epilepsy center between 2022 and 2023. All patients had a minimum follow-up duration of 1 year to assess treatment outcomes. Detailed clinical data for each patient are provided in Table S1. This article was approved by Institutional Review Board of First Affiliated Hospital of Fujian Medical University (number: MRCTA, ECFAH of FMU [2019]274), and we were in accordance with the 1975 Helsinki declaration and its later amendments.

Data acquisition and analysis

SEEG recording

SEEG data were recorded using the Xltek acquisition system (Natus EEG System pk1171 and Neuracle HEEG NSH0256) at sampling rates of 4 or 2 kHz. Electrode placement was determined by the clinical team for each patient. For analysis, a randomly selected interictal snapshot (average duration of 10 min) was used. Interictal sampling was performed at least 2 h apart from seizure events without specific selection criteria, such as the presence or absence of epileptiform activity.

Clinical delineation of epileptogenic zones

The clinical team independently hypothesized the EZ based on non-invasive and invasive data collected during preoperative evaluation. The SOZE was defined as the SEEG channels exhibiting the earliest electrophysiological changes at seizure onset, typically characterized by low-voltage fast activity. The EZ (SOZC) was defined as the anatomical regions requiring ablation or disconnection treatment (RFTC, LITT, surgical resection), including SEEG channels at the seizure onset and those involved in early seizure propagation. Notably, surgical planning was based on the assessment of the EZ, resulting in considerable overlap between SOZC and the actual treatment area.

Clinical classification of surgical outcomes

Surgical outcomes were classified by epilepsy specialists at each center according to the Engel Surgical Outcome Scale and the International League Against Epilepsy (ILAE) classification system. A successful outcome was defined as freedom from disabling seizures for 12 months or longer post-surgery (Engel Class I and ILAE 1–2). Conversely, a failed outcome was defined as the persistence of disabling seizures (Engel Class II–IV and ILAE 3–6) over the same period. Of the 30 patients, 22 achieved seizure freedom (successful outcome), while eight continued to experience seizures (failed outcome). Current clinical experience indicates that MRI-visible lesions are associated with higher success rates, whereas non-lesional, extratemporal, or multifocal epilepsy patients have lower seizure-free rates (Li et al., 1999; Sheikh et al., 2019; Elsharkawy et al., 2008). To categorize clinical complexity (CC), patients were classified as follows: CC I, single lesion (MRI-visible); CC II, mesial or non-mesial temporal lobe epilepsy; CC III, focal or multifocal extratemporal epilepsy; CC IV, multiple neuroimaging abnormalities potentially distributed across multiple brain regions.

Neural fragility network construction

Neural fragility is predicated on the hypothesis that focal epileptic seizures originate from a small subset of fragile nodes, rendering the cortical epilepsy network on the verge of instability. During interictal or preictal periods, the activity recorded from each channel fluctuates around a baseline value. If the network is “balanced,” it responds transiently to perturbations or stimuli, returning to baseline. In contrast, during ictal events, iEEG data demonstrate increased amplitude, oscillations, and propagation across the brain. This study modeled susceptibility to seizure events within the interictal SEEG network. Raw SEEG signals were initially downsampled to 500 Hz and then bandpass-filtered between 0.1 and 30 Hz. After identifying and excluding bad channels, the filtered SEEG signals were analyzed using a modified neural fragility method (Li et al., 2021). Unlike the original neural fragility method, dynamic mode decomposition (DMD) techniques (Brunton et al., 2021) were employed to estimate the linear operator A (state transition matrix), transitioning the state X(t) at time window t to state X(t + 1) at time window t + 1, as described by X(t + 1) = AX(t) + ϵ(t). Here, X(t) represents the SEEG signal within the current time window (window length = 250 ms, overlap = 125 ms), ϵ(t) represents the For this analysis, the Python implementation of DMD (PyDMD v0.4.0.post2301) was utilized with a regularization parameter of 0.05. Subsequently, the minimum 2-norm perturbation matrix for each node in the network was calculated to shift the network from a stable to an unstable state (i.e., triggering a seizure). These minimum perturbations Δ (defined as neural fragility) were then mapped across time and space, generating neural fragility heat maps (y-axis representing channel indices, x-axis representing time). These maps provide a visual representation of the fragility distribution corresponding to the interictal SOZ and other channels, where nodes with high fragility values indicate regions with a greater predisposition to seizures. Fragility heat maps were computed for each individual patient and simulated patient datasets (Fig. 1).

Figure 1 Data collection and processing workflow.

Shows how the vulnerability network we proposed is calculated from a single SEEG data snapshot. Please refer to the content of this section for the descriptions of x, A, and Δ.

Experimental design

Summary analysis of patients

Initially, we analyzed the differences in neural fragility distribution between SOZ and NSOZ (channels outside the SOZ). Data from all patients were aggregated and grouped by surgical outcome (successful vs. failed). To further investigate whether SOZ fragility is significantly higher than NSOZ fragility within different groups, paired t-tests were employed, which is appropriate for comparing two related samples within the same group (e.g., SOZ and NSOZ fragility values in the same patient) to determine if SOZ fragility is significantly elevated. If SOZ fragility is significantly higher than NSOZ in the successful outcome group, it suggests that quantitative analysis of interictal fragility may be valuable for predicting the SOZ. Additionally, we examined the distribution within the seizure snapshots of each patient, depicting the differential distribution of neural fragility between SOZ and NSOZ in individual patients. The hypothesis is that patients with successful outcomes should maintain a higher level of neural fragility in the interictal SOZ.

Validation of interictal fragility features for predicting the epileptogenic zone

To further explore the utility of fragility values derived from interictal SEEG data in localizing the SOZ, we integrated the spatiotemporal fragility heat maps generated from the interictal SEEG snapshots of each patient with a random forest model to establish a binary classification prediction model (SOZ/NSOZ) for individual channels. An effective biomarker should show high concordance with treatment targets in patients with favorable outcomes. Therefore, using the random forest model, we performed binary classification of each patient’s electrode channels (1-SOZ; 0-NSOZ) and compared these results with the clinically identified SOZE and the treatment target SOZC, calculating model prediction precision and recall rates. Subsequently, we constructed a logistic regression model incorporating treatment modality, MRI findings, CC classification, and surgical outcomes to validate the predictive efficacy of fragility features. This involved: (i) performing comparative analyses to validate the predictive performance of neural fragility against clinically identified SOZE (based on EEG) in different outcome groups; (ii) comparing the concordance between model-predicted epileptogenic zones and clinical treatment targets (SOZC) across different surgical outcomes, with further elucidation of the clinical significance of fragility features through specific cases; (iii) analyzing the impact of other clinical variables on the predictive accuracy of fragility features.

Results

The interictal SOZ exhibits significantly greater fragility in patients with favorable surgical outcomes

In a cohort of 30 patients, 22 were classified in the successful outcome group (Engel I), while eight were in the failed outcome group (Engel II–IV). Figure 2A visually depicts the distribution of fragility differences between the SOZE and NSOZE in both the successful (Outcome = 1) and failed (Outcome = 0) groups. During interictal period, electrodes corresponding to the SOZE demonstrated a higher level of fragility compared to those associated with the NSOZE. Figure 2B illustrates that, during the interictal period, the fragility of the SOZ is significantly higher than that of the NSOZ across opposite outcome groups, with a more statistically significant in the successful group (for detailed paired t-test results, refer to Table S2B; successful outcome group: t-statistic = 6.17, p-value = 4.03e−06; unsuccessful outcome group: t-statistic = 2.82, p-value = 0.026).

Figure 2 Distribution of fragility in SOZ and NSOZ electrodes among patients with successful and failed surgical outcomes.

(A) The average fragility distribution of channels corresponding to the SOZ and NSOZ in both successful and failed surgical outcome groups. Electrodes corresponding to the SOZ (red) exhibit a higher average fragility distribution than those corresponding to the NSOZ (purple). (B) In the group with failed outcomes, the fragility of the SOZ (p = 4.03E−06) is significantly higher than that of the NSOZ (p = 0.03). In the group with successful outcomes, the SOZ also shows significantly greater fragility than the NSOZ, with a more pronounced difference than in patients with failed surgery. The average fragility data for each patient’s SOZE/NSOZE and grouped statistical results can be found in Tables S2A and S2B. The p-value represents the probability that the observed difference occurred by chance, with values below 0.05 indicating statistical significance. Abbreviations: SOZE seizure onset zone defined as the SEEG channels exhibiting the earliest electrophysiological changes at seizure onset, NSOZE extra SOZE.

Interictal fragility network for predicting the epileptogenic zone and identifying effective therapeutic targets

We developed a predictive model to classify individual channels by integrating spatiotemporal fragility heatmaps, generated from interictal SEEG snapshots of 30 patients, with a random forest classifier (see Supplemental 3 for details). To determine whether spatiotemporal fragility heatmaps are effective features for predicting the SOZ, we further validated the model against patient outcomes (refer to Table S3 for the prediction results of SOZE and SOZC for each patient).

An optimal predictive model for SOZ should fulfill the following criteria: (i) exhibit a high degree of concordance with SOZE; (ii) demonstrate a higher concordance with the therapeutic target SOZC in patients with favorable treatment outcomes, and conversely, display lower concordance in patients with less favorable surgical outcomes; (iii) show minimal influence from other clinical variables.

High concordance between model predictions and SOZE

Figure 3 illustrates the concordance between the fragility-predicted SOZ and the clinically annotated SOZE (identified as the most likely EZ based on EEG interpretation) across different surgical outcome groups. In the successful outcome group (Engel score = 1), the model achieved an average precision of 0.69 and a recall of 0.93 for SOZE. Conversely, in the unsuccessful outcome group, the model demonstrated a precision of 0.86 and a recall of 0.99. Statistical analysis using t-tests indicated that the difference in precision between the two groups was not statistically significant (p = 0.219), while the difference in recall approached statistical significance (p = 0.057).

Figure 3 Model prediction for SOZE in patients with different outcome.

The box plot presents the distribution of the prediction of fragility model for SOZE (recall and precision) among different outcome groups (Successful Outcome Group: Engel score = 1; Failed Outcome Group: Engel score > 1). The results show that the model achieves a high and stable SOZE recall in both groups, while the precision demonstrates greater variability, which may influence surgical outcomes. (Quartile statistics for each group are provided in Table S4). Abbreviations: SOZE, SOZ seizure onset zone identified with ictal EEG support.

These findings suggest that the interictal fragility predictive model captures a significant portion of the information utilized in SEEG interpretation. However, this does not necessarily translate into improved surgical outcomes, as successful surgery may also depend on other factors, including the patient’s specific pathological and physiological characteristics, as well as the precision of the surgical procedure.

Model concordance with SOZC across different surgical outcomes

The distribution of the model’s concordance metrics (precision and recall) for predicting the SOZ relative to the clinical therapeutic target SOZC across different Engel score groups is shown in Fig. 4. For patients with an Engel score of 1 (successful surgery group), the model achieved a recall rate for SOZC close to 100%. The precision distribution for SOZC was also relatively high (median = 1.00). In patients with an Engel score of 2, although data were limited, both metrics (SOZC recall and precision) were relatively high, with quartiles for recall at 0.97 and precision at 0.86. In the Engel score 3 group (moderate surgical outcome), the median recall rate was 0.8793, and the precision distribution was concentrated around 1.00, indicating that the model performed well in identifying SOZC but with some variability. For patients with an Engel score of 4 (failed surgical outcome), the recall rate for SOZC showed considerable variability, with a minimum value of 0.13 and quartiles ranging from 0.44 to 1.00. This suggests that in patients with failed surgical outcomes, the model’s recall rate for SOZC varied significantly, with lower recall rates observed in some patients. Similarly, precision ranged from 0.24 to 1.00, indicating a high degree of variability in the model’s precision for SOZC in the failed surgery group.

Figure 4 Model prediction for SOZC in patients with different outcome.

The box plot illustrates the performance of fragility prediction in terms of precision and recall with respect to the clinical therapeutic target (SOZC) across different Engel score groups. Patients with an Engel score of 1, indicating optimal surgical outcomes (seizure-free or nearly seizure-free), exhibited relatively high precision, with a median close to 1 and a narrow interquartile range (IQR). This suggests a strong concordance between the fragility predictions and the clinical SOZC in this group. Conversely, patients in the Engel score 4 group, representing the poorest outcomes, demonstrated the lowest median precision and recall, with the greatest variability. This indicates that the model has difficulty matching the clinical target in these cases. Across different outcome groups, the model consistently achieved high precision, with better prognoses (Engel scores closer to 1) associated with higher precision, implying more accurate predictions and a lower false positive rate. Groups with better prognoses also showed higher recall, indicating the model’s enhanced ability to predict effective therapeutic targets in these cases. (Quartile statistics for each group are provided in Table S5). Abbreviations: SOZC, SOZ seizure onset zone identified with clinical RFTC/LITT/Surgery.

Overall, the model demonstrated high precision across different prognostic groups, with higher precision in groups with better outcomes (Engel scores closer to 1), indicating more accurate predictions and a lower false-positive rate. Higher recall rates were also observed in groups with better outcomes, suggesting that the model effectively predicts therapeutic targets in patients with more favorable prognoses.

Interictal fragility as a reliable predictor of clinical treatment outcomes

Among the 30 patients, 11 underwent resective surgery, while 19 did not receive resection and were treated with either RFTC/LITT ablation or neurostimulation therapy. In patients with successful outcomes (12/19, Engel class 1), the model demonstrated a significantly higher concordance between the predicted SOZ and the therapeutic target compared to those with failed outcomes (7/19, Engel classes 2–4). Figure 5A illustrates the relationship between the concordance metrics (recall and precision) of the model-predicted SOZ with SOZE and SOZC and the treatment outcomes (1 = success, 0 = failure). A clear positive correlation can be observed, where higher concordance metrics between the model-predicted SOZ and the clinical therapeutic target SOZC are associated with a higher likelihood of successful surgery/RFTC outcomes.

Figure 5 Association between model-predicted SOZ concordance metrics and treatment outcomes.

(A) The scatter plot illustrates the relationship between the concordance metrics (recall and precision) of the model-predicted SOZ with SOZE and SOZC, and the treatment outcomes (1 = success, 0 = failure). It can be observed that higher concordance metrics between the model-predicted SOZ and the clinical therapeutic target SOZC are positively correlated with the likelihood of successful surgery/RFTC treatment. (B) Displays the regression coefficients and their 95% confidence intervals for each variable in the regression model, indicating each variable’s impact and significance. The confidence intervals for SOZC recall and SOZC precision lie below zero, suggesting that these variables have statistically significant negative coefficients, indicating that higher recall and precision are associated with better surgical outcomes (lower Engel scores). The confidence intervals for RFTC and Surgery are above zero, indicating statistically significant positive coefficients, implying an association with higher Engel scores (poorer surgical outcomes). The confidence intervals for SOZE recall and SOZE precision cross zero, indicating that their influence on surgical outcomes is not statistically significant. Similarly, the confidence intervals for MRI and CC classification also cross zero, suggesting that these variables do not have a significant impact on surgical outcomes within this model. (For detailed regression coefficients, p-values, and 95% confidence intervals, refer to Table S6). Note that the dots (yellow) represent the estimated coefficients for each variable, and the red error bars denote the 95% confidence intervals of these estimates. The horizontal dashed line at zero assists in identifying which coefficients are significantly different from zero. If a coefficient’s confidence interval does not cross the zero line, it indicates that the coefficient has a statistically significant impact on the outcome. Confidence intervals that include zero suggest no significant impact on the outcome. Abbreviations: SOZE recall, SOZ prediction model recall for SOZE; SOZE precision, SOZ prediction model precision for SOZE, SOZC recall, SOZ prediction model recall for SOZC; SOZC precision, SOZ prediction model precision for SOZC; SOZE, Seizure onset zone identified with ictal EEG support; SOZC, Seizure onset zone included in RFTC/LITT/Surgery targets; RFTC, Radiofrequency thermocoagulation; MRI, Magnetic Resonance Imaging; CC, Clinical Complexity.

Logistic regression analysis identified the model’s recall (SOZC recall: regression coefficient = −2.66, p = 0.001) and precision (SOZC precision: regression coefficient = −3.51, p = 0.002) for the therapeutic target as two significant predictors of surgical outcomes (Fig. 5B). These metrics correspond to the model’s ability to capture the actual therapeutic target and the accuracy of predicting the therapeutic target, respectively. The recall and precision for SOZC were significantly negatively correlated with the Engel score, indicating that a higher concordance between the model-predicted SOZ and the therapeutic target SOZC is associated with a lower Engel score (higher surgical success rate). This suggests that the accurate identification of the therapeutic target by the model (high recall and balanced precision) is a key driver of surgical success. RFTC (regression coefficient = 1.20, p-value = 0.03) and the type of Surgery (regression coefficient = 0.66, p-value = 0.04) had a significant positive impact on the Engel score, indicating that different treatment modalities may influence surgical outcomes. While recallE and precisionE were positively correlated with the Engel score, their p-values were greater than 0.05, indicating no significant statistical relevance. MRI and CC treatment modalities did not have a significant impact on the prognostic outcome (p-values > 0.05), suggesting that these variables have a weaker influence on clinical prognosis.

Clinical significance of the interictal fragility prediction model in patients with very low seizure frequency

We have demonstrated that the distribution of fragility differences between interictal SOZ and NSOZ remains relatively stable. The interictal SEEG neural fragility network prediction model shows strong performance in predicting SOZ and clinical outcomes. Here, we present two patients who underwent SEEG monitoring for more than 1 month but did not exhibit seizures (or only showed epileptiform discharges on EEG without typical clinical manifestations). Both patients have been seizure-free for over a year following treatment, confirming the effectiveness of the clinical targets. The interictal fragility prediction model provided satisfactory predictions for effective treatment targets (see Patient 3 in Fig. 6A and Patient 8 in Fig. 6B).

Figure 6 Effective prediction of SOZE and SOZC in patients with low-frequency epileptic seizures.

(A) Illustrates Case 1 (Patient 3), showing interictal EEG recordings (capturing epileptiform discharges without clinical symptoms), an interictal neural fragility heatmap, and a channel matching diagram. In the top three panels, the red bars indicate the fragility-predicted SOZ, the yellow bars represent SOZE, and the blue bars denote SOZC. The preoperative MRI highlights the lesion (red triangle), while the postoperative MRI images reveal the treatment target lesion (yellow triangle). Throughout more than 1 month of SEEG monitoring, the patient did not experience habitual seizures; SEEG detected two epileptiform discharges without clinical manifestations. Clinical treatment involved RFTC lesioning of multiple foci in the right frontal lobe, guided by EEG and imaging findings. The patient has remained seizure-free for 2 years. The fragility heatmap, derived from interictal SEEG data snapshots in this case, demonstrated that the model’s predicted SOZ (red bars) partially matched the clinically identified SOZE (yellow bars) with a precision of 0.58 and a recall of 1.0 (see Table S3C). It completely matched the treatment target SOZC (blue bars), as determined by electroclinical evidence and confirmed by successful surgical outcomes, with both precision and recall of 1.0 (see Table S3C). (B) Presents Case 2 (Patient 8), displaying interictal EEG recordings (showing no epileptiform discharges but indicating abnormal spikes), an interictal neural fragility heatmap, and a channel matching diagram. Again, in the top three panels, the red bars indicate the fragility-predicted SOZ, the yellow bars represent SOZE, and the blue bars denote SOZC. The preoperative MRI was negative, but the postoperative MRI images show the treatment target lesion (yellow triangle). During 1 month of SEEG monitoring and subsequent multiple electrical stimulations, no habitual seizures were recorded. Interictal SEEG data indicated frequent abnormal discharges in the parietal region channels, which were selected as treatment targets for RFTC. The patient has been seizure-free for over one year. The fragility heatmap, generated from interictal SEEG data snapshots in this case (C), exhibited high recall for SOZE (recall = 1.0) and strong concordance with the validated effective treatment targets, achieving a precision of 0.95 and a recall of 1.0 (see Table S3C). Abbreviations: SOZE, Seizure onset zone identified with ictal EEG support; SOZC, Seizure onset zone included in RFTC/LITT/Surgery targets; RFTC, Radiofrequency thermocoagulation; MRI, Magnetic Resonance Imaging; CC, Clinical Complexity.

Discussion

This study systematically explores interictal neural fragility as a potential biomarker for predicting the seizure onset zone (SOZ) and clinical treatment outcomes in patients with drug-resistant epilepsy (DRE). Our findings indicate that patients who underwent successful surgical treatment exhibited significantly higher fragility in the SOZ during the interictal period. By analyzing neural fragility heatmaps generated from interictal SEEG data, we were able to accurately localize the SOZ in both temporal and spatial dimensions, demonstrating high sensitivity and specificity in SOZ prediction. By incorporating Dynamic Mode Decomposition (DMD) techniques, we successfully quantified the susceptibility of network nodes to transition from a stable to an unstable state, thereby identifying regions with a higher potential risk of seizure onset.

Previous studies have primarily focused on the role of ictal neural fragility in SOZ prediction (Burns et al., 2014; Yaffe et al., 2015; Bernasconi, 2017; Li et al., 2017), with less emphasis on interictal data. Our research further expands this field by utilizing interictal SEEG data to provide a novel method for localizing the SOZ without the need for capturing multiple seizure events. This approach is particularly useful for patients with low seizure frequency who are not suitable for prolonged intracranial monitoring. Additionally, our study underscores the characterization of epilepsy as a network disorder, validating that interictal fragility can offer unique insights into the mechanisms of seizure genesis (Bancaud et al., 1970; Jiang et al., 2022; Singh, Sandy & Wiebe, 2015; Breiman, 2001; Lemaître, Nogueira & Aridas, 2017; Chao, Andy & Leo, 2004).

The results of this study hold significant clinical implications. Firstly, by utilizing interictal SEEG data, clinicians can more accurately identify the SOZ and predict treatment outcomes, thereby reducing the reliance on long-term intracranial monitoring, especially in patients with very low seizure frequency during monitoring. This approach not only reduces the risk of infections and complications but also enhances the quality of patient care. Secondly, the correlation between the model and treatment targets suggests that it can aid in identifying patients who are most likely to benefit from specific therapeutic strategies, optimizing treatment selection and improving overall efficacy. For clinical decision-making, this model provides a quantitative and objective tool that aids in more precise surgical planning.

The SOZ prediction model constructed based on interictal neural fragility networks demonstrates high recall and precision rates, particularly in patients with favorable surgical outcomes. This indicates that the model can comprehensively detect potential seizure onset zones, providing more reliable results compared to traditional EEG interpretations. Traditional EEG interpretation often relies on the clinician’s experience and subjective judgment, which may lead to the omission of some potential SOZEs. In contrast, the data-driven interictal fragility model can extract complex features from EEG signals that are difficult to discern through conventional interpretation, offering clinicians a more objective and accurate diagnostic tool.

Despite the significant progress made in this study, several limitations remain. Firstly, the sample size is relatively small and originates from a single clinical center, which may restrict the generalizability of the findings. Although statistical significance was achieved in multiple analyses, a larger, multicenter cohort is essential to validate the robustness and applicability of the interictal fragility model across diverse epilepsy subtypes and clinical settings. Secondly, this was a retrospective study, and as such, the clinical utility of the model could not be evaluated in real-time. Future prospective investigations should incorporate the interictal fragility model into clinical workflows to assess its practicality and clinical value during pre-surgical evaluation. Thirdly, while gamma-band activity was excluded in the present analysis, future research could explore multi-scale fragility models that combine low-frequency network dynamics with high-frequency oscillations, such as ripples (80–250 Hz), to enhance the precision of SOZ localization. Additionally, although the model demonstrated strong concordance with RFTC targets, further anatomical validation in resective surgery cases will require high-resolution postoperative imaging and voxel-wise analyses of resection cavities. Lastly, integrating complementary diagnostic tools such as advanced neuroimaging techniques—including functional MRI, PET, and MEG—may further improve SOZ localization accuracy and enhance the interpretability of fragility-based predictions. This multimodal approach could provide a more comprehensive understanding of the epileptogenic network and support more personalized treatment strategies. Our ongoing prospective study aims to address these limitations and further refine the clinical applicability of this model.

While the fragility-based SOZ prediction model demonstrated strong sensitivity and specificity, several potential confounding factors must be acknowledged. First, inter-patient variability—including differences in underlying pathology, seizure semiology, and clinical complexity—may influence the model’s performance. Second, variability in electrode implantation strategies, such as spatial coverage and targeting precision, can affect the detection of fragile nodes. Additionally, anatomical variability in brain structure among patients may contribute to inconsistencies in model prediction. Future work should aim to systematically assess these factors and evaluate their potential impact on model generalizability and robustness across diverse clinical settings.

Conclusion

In conclusion, this study demonstrates the potential value of interictal neural fragility in predicting SOZ and guiding clinical treatment in patients with DRE. The developed predictive model shows a high degree of concordance with clinical SOZ and treatment outcomes, highlighting its importance in improving epilepsy care. This model provides a new perspective for the quantitative analysis of clinical EEG data and offers a novel diagnostic tool for patients with low seizure frequency, potentially improving diagnostic efficiency and reducing healthcare costs. Future research should focus on further validating and optimizing this model and exploring its potential in personalized treatment strategies.

Supplemental Information

Supplemental Information 1 STROBE checklist.

Supplemental Information 2 Supplementary Materials.

Miao Cao acknowledges the facilities and scientific and technical assistance of the National Imaging Facility, a National Collaborative Research Infrastructure Strategy (NCRIS) capability, at the Swinburne Neuroimaging Facility, Swinburne University of Technology.

Abbreviations

SOZ(A) SOZ seizure onset zone identified with ictal EEG support

SOZ(B) SOZ seizure onset zone identified with clinical RFTC

SOZ(C) SOZ seizure onset zone random forest classifier of fragility predicted

EZ Epileptogenic zone RFTC Radiofrequency thermocoagulation

Additional Information and Declarations

Competing Interests

The authors declare that they have no competing interests.

Author Contributions

Yue Pang conceived and designed the experiments, performed the experiments, analyzed the data, prepared figures and/or tables, and approved the final draft.

Yixuan Yang conceived and designed the experiments, performed the experiments, analyzed the data, prepared figures and/or tables, and approved the final draft.

Yunqing Lin conceived and designed the experiments, analyzed the data, prepared figures and/or tables, and approved the final draft.

Jianyu Zhu conceived and designed the experiments, prepared figures and/or tables, and approved the final draft.

Penghui Liu conceived and designed the experiments, prepared figures and/or tables, and approved the final draft.

Yu Tian conceived and designed the experiments, prepared figures and/or tables, and approved the final draft.

Feng Wang conceived and designed the experiments, analyzed the data, prepared figures and/or tables, and approved the final draft.

Zhen Mei performed the experiments, analyzed the data, prepared figures and/or tables, and approved the final draft.

Dezhi Kang performed the experiments, prepared figures and/or tables, and approved the final draft.

Miao Cao performed the experiments, authored or reviewed drafts of the article, and approved the final draft.

Yuanxiang Lin performed the experiments, authored or reviewed drafts of the article, and approved the final draft.

Human Ethics

The following information was supplied relating to ethical approvals (i.e., approving body and any reference numbers):

This article was approved by Institutional Review Board of First Affiliated Hospital of Fujian Medical University (number: MRCTA, ECFAH of FMU [2019]274), and we were in accordance with the 1975 Helsinki declaration and its later amendments.

Data Availability

The following information was supplied regarding data availability:

The data of the study are available at figshare: Pang, Yue (2024). fragility for seizure. figshare. Dataset. https://doi.org/10.6084/m9.figshare.27641199.v2.

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
