# Peer review of "Interictal neural fragility predicts seizure onset zone and surgical outcomes in drug-resistant epilepsy"

_PeerJ, doi:10.7717/peerj.19548_

## Round 0.1 · original submission · Major Revisions

Dear authors,

Thank you for your submission. Please, refer to the reviewers' comments for further details on required revisions and/or rebuttal.

·

Basic reporting

This paper is structurally clear and easy to follow.

However, the quality of the figures needs improvement. Figure 1 hardly provides any useful information. Even after downloading the separately uploaded dataset, it remains difficult to interpret.

It seems that Figure 5 uses the same statistical results as Figures 3 and 4. If so, please specify this in the paper.

Experimental design

In Section 2.2, "Clinical Marking of the Epileptogenic Zones," I suggest modifying the title such as "Clinical Delineation of Epileptogenic Zones" for clarity.

In line 200, why do the authors focus on data bandpass-filtered between 0.1 and 30 Hz? A cutoff at 30 Hz seems too low, as gamma oscillations also play a role during resting states in the interictal period. Could you explain the reasoning behind this choice?

In line 204, regarding the linear dynamics, I suggest adding a noise term on the right side to improve the model's accuracy.

Validity of the findings

Additionally, it appears that the authors have mixed the definitions of SOZ and EZ. SOZ is not only defined by SEEG channels but also by anatomical regions.
-- To clarify, Rosenow and Lüders (Brain, 2001) defined the SOZ as "the area of the cortex from which clinical seizures are actually generated," whereas the EZ is "the area of the cortex that is indispensable for the generation of epileptic seizures." Subsequently, the concept of EZ networks has been further discussed (Bartolomei et al., Epilepsia, 2017). More recently, modeling approaches have been used to estimate the EZ network by incorporating underlying structural connectivity (Wang et al., Science Translational Medicine, 2023). The authors may also consider describing how Fragility accounts for functional network effects.

I suggest that the authors compare the results with resected regions and surgical outcomes.

Reviewer 2 ·

Basic reporting

The language used throughout the manuscript is clear and unambiguous, which aids in the comprehension of complex subject matter related to epilepsy and treatment outcomes.The literature references are relevant and well-integrated into the text, providing a sufficient background to the field but need some update in reference. The professional structure of the article aligns with the standards of peer-reviewed journals. Figures and tables are appropriately used throughout the manuscript, enhancing the presentation of data. The manuscript appears self-contained, presenting relevant results that support the stated hypotheses.

Experimental design

the experimental design is robust and effectively aligned with the journal's aims and scope. The focus on a well-defined and meaningful research question, along with a commitment to high ethical standards and replicable methods, contributes to the strength of the manuscript.

but some suggestions could further enhance its robustness. Firstly, while the study utilizes a relatively small sample size of 30 patients, author must discuss about the impact in the result and suggest future research. Lastly, incorporating complementary diagnostic techniques, such as advanced neuroimaging modalities, may enhance the accuracy of SOZ localization and provide more comprehensive insights into the predictive value of the fragility model.

Validity of the findings

The validity of the findings in this study is commendable, particularly regarding the strong sensitivity and specificity of the fragility-based SOZ prediction model. Furthermore, a detailed discussion on the impact of potential confounding factors, such as variations in electrode placements or individual patient characteristics, could enhance the rigor of the conclusions drawn.

Additional comments

The manuscript is well-structured, and the language is generally clear, although minor improvements in grammatical precision could enhance readability.

Annotated reviews are not available for download in order to protect the identity of reviewers who chose to remain anonymous.

---

## Round 0.2 · accepted · Accept

Dear authors.
the reviewers are satisfied with the revisions provided. I am now accepting your work for publication, under the condition that you provide figures with better resolution as per production will instruct and is on the authors guidelines. It is particularly important for figures 1, 5 and 6. In figure 6 you also should scales (to tomography images themselves). I also highlight the careful proofreading of the rest of the manuscript. Many thanks for submitting to PeerJ.

·

Basic reporting

No comment

Experimental design

no comment

Validity of the findings

no comment

Additional comments

Thank you — the authors have addressed all my comments.